# The Influence of Body Mass Index on Glucocorticoid Insensitivity in Chronic Rhinosinusitis with Nasal Polyps

**DOI:** 10.3390/jpm12111935

**Published:** 2022-11-21

**Authors:** Yuling Zhang, Shen Shen, Yating Liu, Zaichuan Wang, Qiqi Wang, Yan Li, Chengshuo Wang, Feng Lan, Luo Zhang

**Affiliations:** 1Department of Otolaryngology, Head and Neck Surgery, Beijing Tongren Hospital, Capital Medical University, Beijing 100730, China; 2Beijing Key Laboratory of Nasal Disease, Beijing Institute of Otolaryngology, Beijing 100005, China; 3Department of Allergy, Beijing Tongren Hospital, Capital Medical University, Beijing 100730, China

**Keywords:** body mass index, glucocorticoids, chronic rhinosinusitis with nasal polyps

## Abstract

Background: Reasons for glucocorticoid (GC) insensitivity in chronic rhinosinusitis with nasal polyps (CRSwNP) are not completely clear. Here, we investigate the influence of body mass index (BMI) on GC insensitivity in eosinophilic CRSwNP (eosCRSwNP) and noneosinophilic CRSwNP (noneosCRSwNP) patients. Methods: We recruited 699 CRSwNP patients and gave them a course of oral methylprednisolone for 2 weeks (24 mg/day). Patient demographics and clinical features were analyzed in both GC-sensitive and GC-insensitive CRSwNP patients with different BMI levels and phenotypes. Results: 35.3% of recruited CRSwNP patients were GC-insensitive, and the majority of GC-insensitive patients were males or prone to overweight & obese. Logistic regression analysis further confirmed that being overweight & obese was an independent risk factor for GC-insensitive of CRSwNP patients (odds ratio = 1.584, *p* = 0.049). Compared to underweight & normal-weight patients, overweight & obese patients were more likely to be GC insensitivity, particularly in the eosCRSwNP group, but not in the noneosCRSwNP group. However, there was no significant difference between the underweight & normal weight and the overweight & obese GC-insensitive eosCRSwNP patients regarding the number of infiltrated eosinophils, neutrophils, and polyp recurrence rate. Conclusions: Collectively, our findings demonstrate for the first time that BMI contributes to GC insensitivity in eosCRSwNP patients.

## 1. Introduction

Chronic rhinosinusitis (CRS) is the chronic inflammation of the nasal cavity and paranasal sinuses and can be divided into two phenotypes, CRS with nasal polyps (CRSwNP) and CRS without nasal polyps (CRSsNP) [1,2]. Based on tissue eosinophils and eosinophil-related parameters, CRSwNP is further classified into eosinophilic CRSwNP (eosCRSwNP) and noneosinophilic CRSwNP (noneosCRSwNP) [3]. Although intranasal glucocorticoid (GC) acts as a first-line treatment for CRSwNP patients to reduce the size of nasal polyp tissue and improve nasal symptoms, the response of CRSwNP patients to GC varies widely due to disease heterogeneity [4]. Generally, eosCRSwNP patients are GC-sensitive populations, while noneosCRSwNP patients are more likely to have GC insensitivity [5]. However, factors that influence a patient’s ultimate biologic response to GC are still not fully understood.

Environmental factors such as cigarette smoking and respiratory infections have been shown to impair glucocorticoid receptor (GCR)-α nuclear translocation, which turns out to be a low expression of GCR-α in the airways contributing to GC insensitivity [6,7,8]. With improved standards of living, the incidence of obesity has recently increased remarkably all over the world, especially in China [9]. In overweight and obese asthmatic patients, elevated body mass index (BMI) leads to neutrophil infiltrations [10] and is also associated with *in vitro* dexamethasone blunted response [11]. However, whether BMI level affects GC insensitivity in CRSwNP patients remains to be assessed.

In this study, we recruited CRSwNP patients and collected their characteristics. We further analyzed the effect of BMI levels on GC insensitivity in both eosCRSwNP and noneosCRSwNP patients.

## 2. Materials and Methods

### 2.1. Subjects

We recruited 699 CRSwNP patients for this study from 2010 to 2019 from the Department of Rhinology, Beijing TongRen Hospital (Figure 1). The diagnosis of CRSwNP was made according to the European position paper on rhinosinusitis and nasal polyps 2007 (EPOS 2007) guidelines [12], and patients were excluded if they had unilateral disease, allergic fungal rhinosinusitis, antrochoanal polyps, or cysts. Each patient underwent bilateral, standardized, and diagnostic rigid nasal endoscopy by two rhinologists. Patients were blindly scored according to the Lund–Kennedy scoring system [13]. To better access GC response, this study excluded the patients whose Lund–Kennedy score was less than 4. Patients who were less than 18 years old, pregnant, or with uncontrolled underlying disease or severe infection were also excluded. None of the patients had taken oral GC, antibiotics, or other immunomodulatory drugs within 4 weeks before the recruitment. This study was approved by the Ethics Committee of Beijing TongRen Hospital, and all subjects signed informed consent forms before enrollment in the study.

### 2.2. Clinical Feature Collection

Patients’ BMIs (body weight in kilograms divided by the square of height in meters) were classified into four categories according to World Health Organization (WHO) reference standards: underweight (BMI < 18.5 kg/m^2^), normal weight (18.5 ≤ BMI < 25 kg/m^2^), overweight (25 ≤ BMI < 30 kg/m^2^) and obese (BMI ≥ 30 kg/m^2^) [14]. Due to the limited number of underweight and obese patients (respectively 21 and 47 patients), we incorporated the patients, and finally, we have two groups, the underweight & normal weight group, and the overweight & obese group.

Visual analog scale (VAS) scores for total and individual sinonasal symptoms were marked by the patients on a 10-cm scale, with 0 meaning complete absence of symptom and 10 being the worst thinkable severity. Patients were asked to score each individual symptom (rhinorrhea, nasal obstruction, reduction in the sense of smell, and headache) on a scale from 0 to 10. Medical history information of CRSwNP patients, including asthma comorbidity and allergy, as defined by the Global Initiative for Asthma 2006 guidelines [15] and the Allergic Rhinitis and its Impact on Asthma 2008 criteria, respectively [16]. Computed tomography (CT) scans were performed on enrollment, and affected sinuses were graded according to the Lund–Mackay staging system [17]. Frontal, sphenoid, maxillary, and anterior and posterior ethmoid sinuses at each side were graded between 0 and 2 (0: no abnormality, 1: partial opacification, 2: total opacification), and the ostiomeatal complex was scored as 0 (not obstructed) or 2 (obstructed).

### 2.3. Clinical Phenotype Evaluation in CRSwNP Tissue

Nasal polyp biopsies of patients were collected in the clinic and paraffin-embedded for further eosCRSwNP and noneosCRSwNP definition. All sections of nasal polyps tissue were stained with hematoxylin and eosin (H&E) and were blindly counted under a light microscope (Olympus, BX51) at 400× magnification by two pathologists. The average counts of eosinophils, neutrophils, plasma cells, and lymphocytes from 3 non-overlapping fields were recorded. Tissue eosinophil absolute counts >55 per high power field (HPF) were defined as eosCRSwNP [18].

### 2.4. Assessment and Follow-Up of GC Treatment Efficacy

After obtaining a nasal polyp biopsy, CRSwNP patients received oral methylprednisolone (24 mg/day for 2 weeks). GC treatment efficacy was assessed on the 15-day post-oral GC treatment. Patients were designated as GC-insensitive when the Lund–Kennedy score of nasal polyps was reduced by less than 1. Otherwise, CRSwNP patients were regarded as GC-sensitive. All patients showed good compliance with oral GC, and no severe comorbidity was reported. After oral GC treatment, patients underwent endoscopic sinus surgery, if necessary, which involved removing nasal polyps and opening up the natural ostia of the diseased sinuses. All patients adhered to the same follow-up schedule and applied with intranasal GC spray, mometasone furoate monohydrate (Nasonex, MSD, Belgium) 100 μg in each nostril once a day, and nasal irrigation therapy if necessary [19]. The recurrence of patients was defined by the presence of nasal polyps observed under the nasal endoscopy and one or more symptoms (nasal blockage, nasal discharge, facial pain, and smell loss) that persisted for at least one week during follow-up [20]. The end time point of the follow-up was September 2020, and the follow-up period was 69.5 ± 30.1 months.

### 2.5. Statistical Analysis

To determine independent risk factors of GC insensitivity in CRSwNP, potential predictors identified by univariate regression model (*p* values < 0.1) were chosen for further multivariate logistic regression analysis. Odds ratios (OR) with a 95% confidence interval (CI) of multivariate logistic regression were reported. Statistical analysis was performed using SPSS 23.0 software (IBM, Armonk, New York, NY, USA) and R software (R Foundation, Vienna, Austria) (version 4.0.3; http://www.r-project.org, assessed on 20 January 2021). Continuous variables were displayed as mean ± standard deviation (SD) and analyzed by Mann–Whitney U tests. Categorical variables were displayed as numbers and percentages and analyzed by the Chi-square test. Differences were considered significant when *p* values < 0.05.

## 3. Results

### 3.1. Being Overweight & Obese Was a Risk Factor for GC Insensitivity in CRSwNP

As shown in Table 1, 35.3% of CRSwNP patients were GC insensitive, while 64.7% of patients were GC sensitive. The majority of GC-insensitive CRSwNP patients were males (78.9%) and predisposed to be overweight & obese (48.2%). In addition, GC-insensitive CRSwNP patients had lower smell loss scores than GC-sensitive CRSwNP patients (4.2 vs. 5.1). Importantly, being overweight & obese was an independent risk factor for GC insensitivity in CRSwNP patients, as confirmed by logistic regression analysis (OR = 1.584, 95% CI = 1.002–2.505, *p* = 0.049) (Figure 2).

### 3.2. Being Overweight & Obese Contributed to GC Insensitivity in eosCRSwNP Patients without Affecting Polyp Recurrence

Considering the risk factor for GC insensitivity, all CRSwNP patients were further classified into four subsets with a combination of BMI levels and clinical phenotypes (Table 2). Regardless of BMI levels, the GC-insensitive patients were predominantly found in the noneosCRSwNP group. Surprisingly, overweight & obese patients were more likely to be GC-insensitive compared to underweight & normal weight patients only in the eosCRSwNP group (9.9% vs. 4.2%, *p* < 0.05). When counting the infiltrated immune cells in tissues, we noticed that there was no significant difference between the underweight & normal weight and the overweight & obese GC-insensitive eosCRSwNP patients (Figure 3A,B). Furthermore, no significant difference in the polyp recurrence rate of GC-insensitive eosCRSwNP patients was observed between the overweight & obese group and the underweight & normal weight group (Figure 3C).

## 4. Discussion

To our knowledge, our study, for the first time, shows that being overweight & obese is an independent risk factor for GC insensitivity in CRSwNP patients. Furthermore, BMI contributes to GC insensitivity in eosCRSwNP patients without the interference of recurrence rate. And the infiltration of neutrophils and eosinophils is not responsible for the effect of BMI on GC insensitivity in eosCRSwNP.

Severe asthma patients are predominantly overweight and obese [21,22]. In the Optimum Patient Care Research Database and British Thoracic Society Difficult Asthma Registry, the obese group exhibited greater asthma medication requirements in terms of maintenance corticosteroid therapy compared to the overweight and the normal weight patients [23]. Similarly, we demonstrate that BMI is the risk factor for GC-insensitive in CRSwNP patients. However, it is noticed that BMI contributes to GC insensitivity only in eosCRSwNP patients rather than noneosCRSwNP. When analyzing the protein level of multiple GR protein isoforms in CRSwNP, our teams found that GCRα-D was increased in noneosCRSwNP compared to eosCRSwNP [24]. Consequently, we assume that BMI contributes to GC insensitivity in eosCRSwNP, not due to the expression of GCRα.

Obesity has also been shown to increase neutrophil infiltration in adipose tissue in the eosinophilic mice model on a high-fat diet [25]. Furthermore, neutrophilic asthma is usually regarded as a severe and poorly controlled asthma subtype [26], and the increased neutrophils in patients with acute or persistent asthma are correlated with a poor response to inhaled corticosteroids [27]. In line with this literature, neutrophilia in CRSwNP patients has also been associated with lower responses to corticosteroid treatment [5]. However, when comparing the infiltration count of neutrophils in nasal polyp tissue of GC-insensitive eosCRSwNP patients with different BMI levels, we did not observe any significant difference. Additionally, our study indicates that being overweight or obese did not affect the response to GC in noneosCRSwNP patients. Therefore, we postulate that neutrophils play a limited role in overweight/obesity-related GC insensitivity in eosCRSwNP patients, and further investigations are needed to clarify.

A nationwide population-based study identified obesity as a risk factor for the prevalence of CRSwNP; whether obesity affects polyp recurrence is still unclear [28]. Our data showed that BMI levels did not affect polyp recurrence rates in GC-insensitive CRSwNP patients. This might be due to the limited polyp recurrence population of GC-insensitive patients. More precise conclusions can be drawn according to the result from a larger cohort. The usage of GC nasal spray after the operation varied among CRSwNP patients as well, which might slightly affect CRSwNP control and polyp recurrence rates.

This study is limited in that the number of obese CRSwNP patients was relatively small due to the characteristics of CRSwNP patients in China. Reducing the effective dose of methylprednisolone pro kilo in obese patients might result in a less-effective therapeutic effect in comparison to underweight patients. Thus, a weight-based dosage of corticosteroids is required in the real world to assess the impact of BMI on corticosteroid treatment. BMI might influence GC insensitivity via the level of obesity-related cytokines, and the underlying mechanisms await further investigations. In particular, obesity-related cytokines are worth testing in nasal polyp tissues in each subgroup from this study in the near future.

## 5. Conclusions

In summary, we found that being overweight or obese contributed to GC insensitivity in eosCRSwNP patients. It also supported the extended attention for prescribing effective doses of corticosteroid according to the weight of eosCRSwNP patients. Whether weight control is beneficial for GC sensitivity in eosCRSwNP requires further and long-term follow-up.

## Figures and Tables

**Figure 1 jpm-12-01935-f001:**
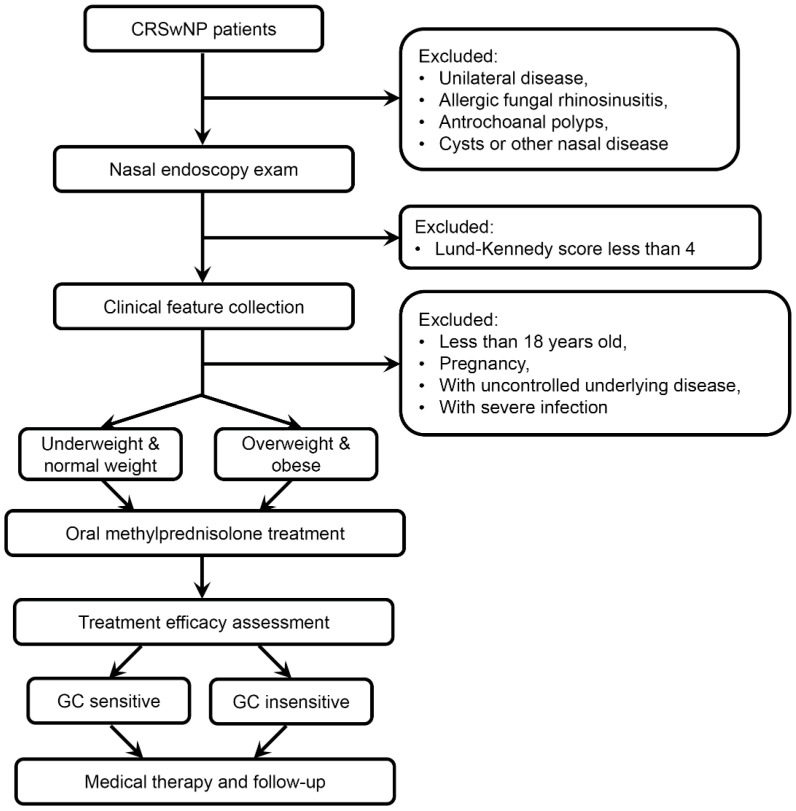
The flow chart of the study design.

**Figure 2 jpm-12-01935-f002:**
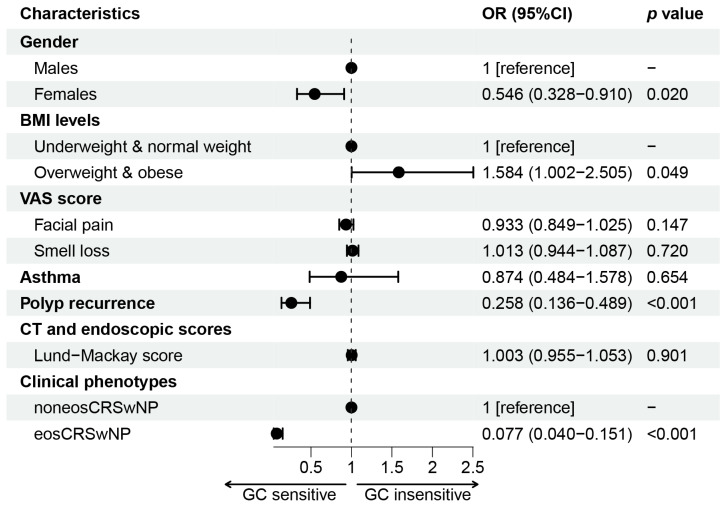
Logistic regression of glucocorticoid (GC) insensitivity. OR: odds ratio; CI: confidence interval; BMI: body mass index; VAS: visual analog scale; NP: nasal polyp; CT: computed tomography.

**Figure 3 jpm-12-01935-f003:**
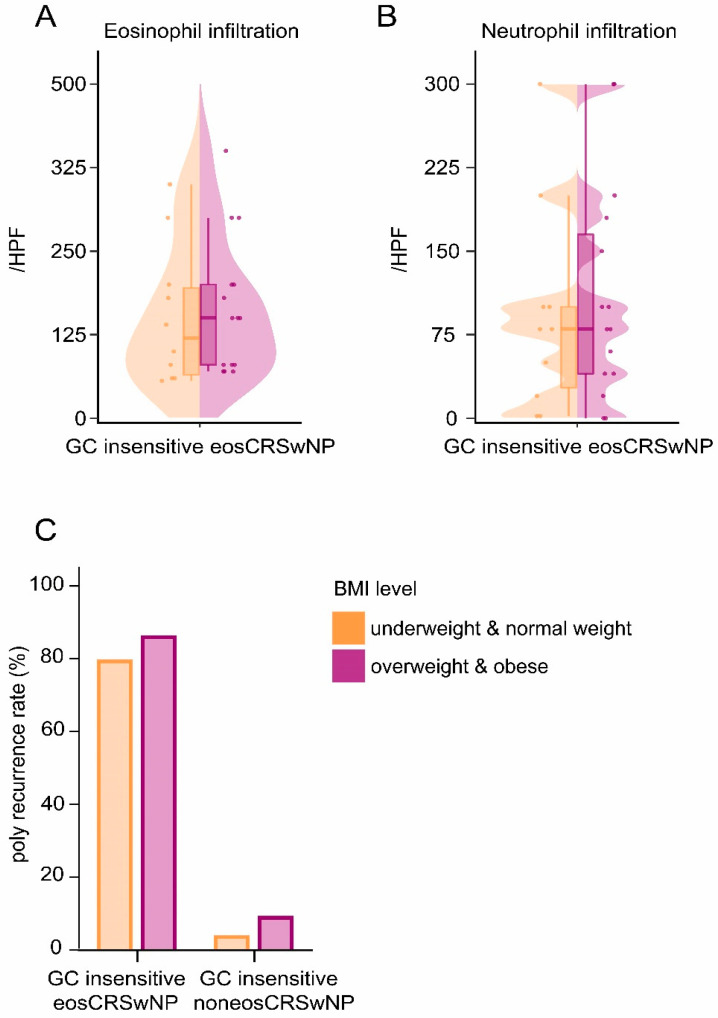
(**A**) Eosinophil infiltration absolute counts in nasal polyp tissues of underweight & normal weight and overweight & obese glucocorticoid (GC)-insensitive eosCRSwNP patients. (**B**) Neutrophil infiltration absolute counts in nasal polyp tissues of underweight & normal weight and overweight & obese GC-insensitive eosCRSwNP patients. (**C**) Polyp recurrence rates of underweight & normal weight and overweight & obese GC-insensitive eosCRSwNP and noneosCRSwNP patients.

**Table 1 jpm-12-01935-t001:** Characters of glucocorticoid (GC)-insensitive and sensitive CRSwNP patients.

Characteristics	GC-Insensitive	GC-Sensitive	*p*-Value
Number	247	452	
Age (mean ± SD)	46.1 ± 13.3	46.1 ± 11.9	0.894
Gender (%)			
Females	52 (21.1)	169 (37.4)	<0.001
Males	195 (78.9)	283 (62.6)	
BMI levels (%)			
Underweight & normal weight	128 (51.8)	282 (62.4)	0.007
Overweight & obese	119 (48.2)	170 (37.6)	
BMI (mean ± SD)	25.1 ± 3.6	24.3 ± 3.5	0.002
Smoke (%)	72 (29.1)	114 (25.2)	0.261
VAS score (mean ± SD)			
Nasal blockage	6.3 ± 2.0	6.2 ± 2.2	0.570
Nasal discharge	5.3 ± 1.9	5.3 ± 2.0	0.629
Facial pain	1.8 ± 2.3	2.2 ± 2.4	0.068
Smell loss	4.2 ± 3.4	5.1 ± 3.3	<0.001
Asthma (%)	32 (13.0)	160 (35.4)	<0.001
Allergy (%)	32 (13.0)	80 (17.7)	0.102
Polyp Recurrence (%)	36 (14.6)	361 (79.9)	<0.001
CT and endoscopic scores (mean ± SD)			
Lund–Kennedy score	6.9 ± 2.0	6.9 ± 1.9	0.695
Lund–Mackay score	17.6 ± 5.0	18.9 ± 4.4	0.002

SD: standard deviation; BMI: body mass index; VAS: visual analog scale; NP: nasal polyp; CT: computed tomography.

**Table 2 jpm-12-01935-t002:** Glucocorticoid (GC) insensitivity rates in each subgroup.

CRSwNP		Number	GC Insensitive (%)	GC Sensitive (%)	*p* Value
eosCRSwNP	Underweight & normal weight	237	10 (4.2)	227 (95.8)	0.027
	Overweight & obese	152	15 (9.9)	137 (90.1)	
noneosCRSwNP	Underweight & normal weight	173	118 (68.2) a	55 (31.8) a	0.135
	Overweight & obese	137	104 (75.9) b	33 (24.1) b	

a—compared to underweight & normal weight eosCRSwNP, *p* < 0.05; b—compared to overweight & obese eosCRSwNP, *p* < 0.05.

## Data Availability

Not applicable.

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
