# Peer review of "The Influence of Body Mass Index on Glucocorticoid Insensitivity in Chronic Rhinosinusitis with Nasal Polyps"

_jpm, 2022, doi:10.3390/jpm12111935_

Round 1

Reviewer 1 Report

Thank you for this chance to review this article entitled The influence of body mass index on glucocorticoid insensitivity in chronic rhinosinusitis with nasal polyps. The work is good and has some scientific merits to be published. However, I would suggest some points to be addressed for the betterment of the scientific quality of the present form.

1. The authors are suggested to get their article edited by a native English Speaker.

2. Figure 1 looks clumsy it need to be clear for better readiability.

3. Clinical feature collection do not looks good. It can be changed.

4. Why did the authors consider significant values at P less than 0.1 for the clinical variables identified by univariate regression models??? Was there any special reasons or explanation?

5. The authors are suggested to draw more valid and robust conclusion based on their findings. 

Reviewer 2 Report

The article is well written and the message these colleagues are trying to convey is clear. Defining prognostic criteria for CRSwNP response to corticosteoroids is foundamental, and it is especially important in time were biological therapy is largely used in this kind of pathology. 

However, this study raises some concerns:

1) Methylprednisone dose used is standard for all patients, independently of patients' BMI. This results in a different dosage pro kilo, reducing effective dose of Methylprednisone pro kilo in obese patients compared to underweight patients. As being overweight as the independent risk factor for GC is at the limits of statistical significance, this aspect needs to be addressed.

2) The definition of EosCRSwNP is unclear (line 74): the authors use 55/hpf as a cut-off, but this cut-off is used in a previous article as an indicator for probability of recurrence of disease, and not as an indicator for eosCRswNP. Lots of guidelines suggest different cut-off for definition of T2I. A recent review "Comparison of guidelines for prescription and follow up of biologics for chronic rhino sinusitis with nasal polyps" by Rampi et al. compares these guidelines, so that the author may choose one that is more fit. 

3) The use of intranasal corticosteroids (line 82) must be specified: which molecule was used and at what dosage and frequency?

Reviewer 3 Report

I congratulate the authors for the innovative and original research on the relationship between BMI and GC resistance in patients with chronic rhinosinusitis with polyposis. In my opinion the paper is worthy of publication

Author Response

Thanks for the review and comments.

Round 2

Reviewer 1 Report

This manuscript can be considered for publication.